# Social Disruption Impairs Predatory Threat Assessment in African Elephants

**DOI:** 10.3390/ani12040495

**Published:** 2022-02-17

**Authors:** Graeme Shannon, Line S. Cordes, Rob Slotow, Cynthia Moss, Karen McComb

**Affiliations:** 1School of Natural Sciences, Bangor University, Bangor LL57 2UW, UK; 2School of Ocean Sciences, Bangor University, Bangor LL59 5AB, UK; l.cordes@bangor.ac.uk; 3School of Life Sciences, University of Kwazulu-Natal, Pietermaritzburg 3209, South Africa; slotow@ukzn.ac.za; 4Amboseli Trust for Elephants, Nairobi 00509, Kenya; cmoss@elephanttrust.org; 5Mammal Communication and Cognition Research Group, School of Psychology, University of Sussex, Brighton BN1 9QH, UK

**Keywords:** playback experiment, *Loxodonta africana*, lions, *Panthera leo*, social structure, anthropogenic disturbance, ecological knowledge, social learning, animal culture

## Abstract

**Simple Summary:**

The sharing of social and ecological information is vitally important for group-living animals, especially among cognitively advanced species (e.g., primates, cetaceans and elephants) that can acquire detailed knowledge over their long lifetimes. In our study, we compared the ability of elephants from two very different populations to assess the threat associated with different numbers of roaring lions. The population in Amboseli (Kenya) consists of stable family groups and experiences relatively low levels of human disturbance, while the population in Pilanesberg (South Africa) was founded in the early 1980’s from young and often unrelated orphan elephants. We broadcast lion roars to families of elephants in both these populations and recorded how they responded to differing levels of threat (one versus three lions). The Amboseli population successfully increased their defensive bunching behaviour to the greater threat associated with three lions, whereas the Pilanesberg elephants appeared unable to make the same distinction. Our findings indicate that profound disruption experienced early in life and the lack of older adults to learn from has impaired the ability of the Pilanesberg elephants to make accurate assessments of predatory threat. We suggest that, in addition to population size, conservation practitioners need to consider the crucial role of social structure and knowledge transmission in these highly social and long-lived species.

**Abstract:**

The transmission of reliable information between individuals is crucial for group-living animals. This is particularly the case for cognitively advanced mammals with overlapping generations that acquire detailed social and ecological knowledge over long lifetimes. Here, we directly compare the ecological knowledge of elephants from two populations, with radically different developmental histories, to test whether profound social disruption affects their ability to assess predatory threat. Matriarchs (≤50 years of age) and their family groups received playbacks of three lions versus a single lion roaring. The family groups in the natural Amboseli population (Kenya) reliably assessed the greater predatory threat presented by three lions roaring versus one. However, in the socially disrupted Pilanesberg population (South Africa), no fine-scale distinctions were made between the numbers of roaring lions. Our results suggest that the removal of older and more experienced individuals in highly social species, such as elephants, is likely to impact the acquisition of ecological knowledge by younger group members, particularly through the lack of opportunity for social learning and cultural transmission of knowledge. This is likely to be exacerbated by the trauma experienced by juvenile elephants that witnessed the culling of family members and were translocated to new reserves. With increasing levels of anthropogenic disturbance, it is important that conservation practitioners consider the crucial role that population structure and knowledge transfer plays in the functioning and resilience of highly social and long-lived species.

## 1. Introduction

The effective transmission of information between individuals is crucial for group-living animals, as it enables appropriate behavioural responses to be coordinated in dynamic environments [1,2]. This is particularly apparent among large-brained, social species with overlapping generations that acquire detailed knowledge over long life spans, including primates, elephants and cetaceans (but is also relevant to many social bird species). Older and more experienced individuals in these groups have been shown to make better decisions when it comes to assessing social risk [3], predatory threat [4] and locating food during periods of resource scarcity [5]. The close social relationships among group members allows for crucial knowledge to be effectively transmitted from one generation to the next via social learning, particularly among species with prolonged juvenile and adolescent life stages [6,7]. Ultimately, this can have direct fitness benefits for individual group members through greater reproductive success and increased probability of survival [3,8,9].

Knowledge acquisition and transmission can be specific to a group or population of animals depending upon the nature of the social and environmental challenges they face, which can lead to behavioural variants or cultural differences that are unique to a particular region [10]. For example, African savanna elephants (*Loxodonta africana*) in Amboseli National Park, Kenya, demonstrated the ability to discriminate the threat posed by different human ethnic groups living in the region from language [11] and olfactory cues [12], a skill that must have involved learning the significance of local variants. In Shark Bay, Australia, Indo-Pacific bottlenose dolphins (*Tursiops aduncus*) learnt to use sponges as effective foraging tools, with mothers passing on this skill to their offspring via social learning [13,14]. There has been increasing interest focused on the important role of knowledge and culture in conservation, particularly in terms of how information acquisition and transmission interact with long-term population viability [10,15,16]. These considerations are especially important for large-bodied and long-lived species, such as the African savanna elephant, which must locate crucial resources [17] while also avoiding a myriad of threats associated with natural predators [18,19] and humans [11] across their expansive home ranges. 

Anthropogenic disturbance of elephant populations has increased markedly over recent decades due to accelerating habitat fragmentation [20,21], overharvesting (i.e., illegal and legal killing) [22,23], climate change [24] and intensifying competition for resources, which has led to widespread human–elephant conflict [25]. These impacts extend beyond the immediate and direct loss of individual animals to having a lasting and profound effect on population integrity because of compromised development and impaired social relationships [26]. For example, male African elephants orphaned as a result of culling and translocation operations in the 1970s and 1980s demonstrated a tendency for hyper-aggressive behaviour on reaching sexual maturity [27,28], while females and their family groups exhibited impaired social knowledge decades after these traumatic events [29]. Furthermore, poaching events in East Africa led to the formation of orphan groups of elephants that commonly consist of unrelated individuals, demonstrate weaker social relationships and rarely benefit from associating with older and more experienced individuals [30,31,32]. The mother–offspring bond is particularly important, as demonstrated by the reduced survival probability of orphaned elephant calves who lost their mothers early on in life [33]. 

These studies clearly indicate the potential impacts of anthropogenic disturbance on social knowledge acquisition and transmission, but there has been comparatively little research—in elephants or other long-lived mammals—focused on whether ecological knowledge is affected by social disruption in a similar manner. One facet of ecological knowledge that lends itself to experimental investigation is the ability to accurately assess predatory threat, a key skill that has direct consequences for survival. Female African elephants live in complex fission–fusion societies where the core social unit is the family group [34]. The matriarch (or oldest female) leads the group, playing a central role in decision making and coordinating behavioural responses to perceived threat [3]. Lions (*Panthera leo*) present the main natural predator of African elephants [18,19,35], particularly to vulnerable calves [36,37], and accurately assessing the predation risk is a key skill for matriarchs and their family groups. Indeed, larger groups of lions present a much greater threat than singletons due to their very effective group hunting strategies [18,19].

We previously used playback experiments in Amboseli National Park, Kenya to demonstrate that individuals within a social group can benefit directly from the influence of an older leader because of their enhanced ability to make crucial decisions about predators [4]. While elephant family groups within a stable, natural study population were consistently able to assess the greater risk associated with three roaring lions versus one, those with older matriarchs were significantly better at the more subtle task of identifying the increased threat associated with male versus female roars (male lions being more than 50% larger and at a distinct advantage in capturing large-bodied prey [36,38,39]). However, ecological knowledge may be affected by experience, as well as age, particularly given that many free-ranging elephant populations have been subjected to severe human disturbance through habitat loss, culling and poaching [22].

In this study, we used playback experiments of one versus three lions roaring to directly compare ecological knowledge in two populations of African elephants, which have experienced radically different life histories. The natural, free-ranging population in Amboseli National Park experiences relatively low-level disturbance by anthropogenic activities and has, therefore, retained a largely normal age and social structure (i.e., family groups of related adult females ranging from 15 to 60 years of age and their calves) [34]. In contrast, our study population in the Pilanesberg National Park, South Africa, was founded with orphan elephants translocated following management culls of adult and older juvenile animals in the Kruger National Park during the early 1980s [27]. As well as experiencing profound trauma from the culling of family members and translocation to a new environment, the surviving orphan elephants commonly formed groups with unrelated individuals [29]. If social disruption impacts the acquisition and transmission of ecological knowledge, we predicted that this would impair the ability of Pilanesberg elephants to accurately discriminate between different levels of predatory threat. 

## 2. Materials and Methods

### 2.1. Study Populations

Playback experiments were conducted in Amboseli National Park (ANP), Kenya, and Pilanesberg National Park (PNP), South Africa, between May 2007 and December 2010. Amboseli encompasses 390 km^2^ of predominantly savanna grassland habitats that surround semi-permanent and permanent swamps, which are fed with run-off from the Kilimanjaro Mountain catchment. The Amboseli Elephant Research Project was established in 1972 and has long-term, detailed demographic records on ~1500 individual elephants within the population, which make up 58 distinct family groups [40]. Individuals born after 1972 have been accurately aged, while the ages of the older elephants were estimated using criteria that are accepted as a standard in studies of African elephants, including the length of the hind footprint, dentition, shoulder height and back length [41]. There was an average of 12 (±7 s.d.) elephants in the Amboseli family groups, including a mean of 4 (±2 s.d.) adult females (see Appendix A for further details).

Pilanesberg is a fenced reserve, which was established in 1979 and comprises approximately 550 km^2^ of hilly, savanna terrain. The vegetation is diverse, including grassland habitats, open savanna woodland and dense bush land thickets. A total of 76 young orphan elephants (<10 years old) were introduced into Pilanesberg from the Kruger National Park between 1981 and 1993 following culling operations that targeted adult family members, while two adult females (19 years of age) were introduced in 1982 [42]. By the year 2000, 11 distinct breeding groups had formed (consisting of unrelated individuals, as noted above), each led by a dominant female (Slotow unpublished data). The movements and associations of these elephant groups were studied continuously from January 2000 to September 2007, with data available for the composition of each family group, as well as for estimating the ages of all adult females (Appendix A). In 2010, the elephant population numbered approximately 200 individuals, including 16 distinct breeding groups (hereon referred to as family groups to be consistent with Amboseli). There was an average of 10 (±5 s.d.) elephants in the Pilanesberg family groups, which included a mean of 3 (±2 s.d.) adult females (see Appendix A for further details). 

The density of lions was similar in both protected areas during the period of study (2007–2010). Pilanesberg had a population of 45–50 lions, which equates to 8–9 lions per 100 km^2^ [43], compared with 6–8 lions per 100 km^2^ in Amboseli National Park [44].

### 2.2. Playback Procedure

The ability to discriminate between different levels of predatory threat was tested in both study populations by broadcasting playbacks of three lions verses a single lion roaring to elephant family groups, with the sex of the roaring lions balanced equally across all playbacks following the protocol of McComb et al. [4]. A total of 125 playback experiments broadcasting lion roars to individual elephant family groups (*n* = 55) was conducted in Amboseli and Pilanesberg, with a mean of 2.3 playbacks per family group. Our previous research demonstrated that matriarch age plays a key role in both ecological and social discrimination tasks [3,4]. Therefore, to directly compare the two populations without the potential confounding effects of age, we only used playback experiments for the Amboseli analysis that were drawn from the same age range as the matriarchs of Pilanesberg (≤50 years), resulting in a final total of 93 playbacks (Amboseli *n* = 40, Pilanesberg *n* = 53).

Eight different playback exemplars were used to assess responses to predatory threat. These consisted of two recorded tracks of one versus three lions roaring for both female and male lions (see also: [4]). The territorial lion roars were recorded in the Serengeti National Park, Tanzania, using Sennheiser MKH816T microphones and Panasonic SV-250 Digital Audio Tape Recorders as part of previous studies [45,46]. A single recording of lion roars (average track length: males, 39 ± 7 s; females, 40 ± 5 s) was played back to the target family group through custom-built loudspeakers manufactured by Intersonics Inc., Northbrook, IL, U.S.A., and B&W loudspeakers, Steyning, U.K. The Intersonics loudspeaker was powered by a Kenwood KAC-PS400M amplifier and the B&W loudspeaker by Alpine PDX-1.1000 and MRP-T222 amplifiers. The peak sound pressure level of the playbacks was standardised to 116 dB at a distance of 1 m from the loudspeaker, which is comparable to natural lion roars [4,46]. A CEL-414/3 sound level meter was used to measure sound pressure levels. In all playback experiments, the study vehicle was positioned ~100 m from the edge of the target family group.

The first exemplar played to a family group was randomised; repeat playbacks to the same family group were then matched with the first so that the sex and number of lions were systematically varied across playbacks. A minimum of seven days was left between experiments to the same family to avoid habituation, while playbacks were not broadcast to groups with calves that were less than 1 month old, as our previous research highlighted that the presence of such very young calves may result in abnormally high sensitivity to perceived threats [3]. 

The behavioural responses of the elephants to playback were observed through binoculars and recorded with a video camera (Canon XM2) alongside live commentary [3,4]. Three behavioural responses were detailed from the video records according to the following established criteria [3,4,11,29]: **Bunching**: A defensive response by the matriarch and her family to a perceived threat, which leads to a reduction in the diameter of the group after the playback experiment (measured in elephant body lengths), **Prolonged listening**: Matriarch exhibits a continued listening response for >3 min after the playback is broadcast, where ears are held away from the head in a stiff and extended position, often with the head slightly raised, and **Bunching intensity**: a metric which categorises the speed and cohesion of the bunching response on a 4-point scale: (0) no bunching recorded, (1) a subtle reduction in group diameter, elephants continue with pre-playback behaviours and remain relaxed (bunch formation >3 min), (2) the group bunch in a coordinated manner with the interruption of pre-playback behaviours such as feeding (bunch formation 1–3 min) and (3) the elephants are very alert and undertake a rapid reduction in diameter of the group (bunch formation <1 min).

An independent observer who was blind to the experimental sequence and did not have access to the video commentary second coded a randomly selected sample (15%) of the playback videos. The level of agreement achieved was 96% for bunching and prolonged listening, while the scores for bunching intensity were also highly consistent with a Spearman’s *ρ* correlation of 0.98 (*p* < 0.0001).

### 2.3. Data Analysis

Data from Amboseli and Pilanesberg were analysed separately to quantify the ability of elephants within each population to assess predatory threat. This was repeated for each of the three response variables, namely bunching, prolonged listening and bunching intensity, which resulted in a total of six model sets (three for each elephant population). Each family group in Amboseli (*n* = 25) and Pilanesberg (*n* = 15) received 1–3 (mean = 1.6) and 1–6 (mean = 3.3) playbacks, respectively (see Appendix A). The lme4 package [47] was used to construct binomial generalised linear mixed models (GLMMs) for bunching and prolonged listening, with family group added as a random effect to account for repeated playbacks to the same elephants. As bunching intensity was an ordered categorical variable, cumulative link mixed models (CLMMs) using the ordinal package [48] were employed for this analysis. No simple predict function existed for CLMM, therefore, we estimated the fitted values using a cumulative link model (CLM). 

To explore the ability of elephant matriarchs to discriminate between different levels of predation risk, we constructed six models within each model set (36 models in total). The models were kept simple due to the relatively small datasets, but, even so, the analysis of prolonged listening for Pilanesberg elephants had models with random-effect variance estimates that were (nearly) zero. Model sets included a single-variable model with number of lions (1 versus 3), two variable additive models with number of lions and either age of matriarch or the number of adult females in the group and a three-variable additive model including number of lions, age of matriarch and number of adult females in the group. We also ran a model with the interaction between number of lions and the age of the matriarch, as well as a null model. As the number of lions was included in all models, the null model enabled us to test the explanatory power of this variable. Preliminary analyses conducted during our previous research revealed that the presence of a calf under three years of age had no bearing on the behavioural responses of the matriarch and her family group to varying levels of predatory risk [4]. As such, we did not include this variable in our modelling approach.

For each of the six model sets, the AICc (Akaike information criterion adjusted for small sample size) values and weights were extracted for all candidate models using the modavg package [49]. Model averaging was conducted across models that accounted for ≥0.95 of the AICc weight to extract parameter β-estimates and their 95% confidence intervals (CI). The significance of the results was assessed by whether the 95% CI of the β-estimate overlapped zero. Fitted values were extracted from the top model using the ‘predict’ function from either the lme4 or ordinal package. 

## 3. Results

The overall probability of bunching—irrespective of the number of lions roaring—was higher in Pilanesberg (57%) than Amboseli (40%) when comparing the raw data. Modelling of the binomial bunching response variable revealed that there was one clear top model for Amboseli, which only included the number of lions variable and accounted for 0.85 of the AICc weight (Table 1). Model averaging revealed a significant difference in the probability of bunching as a function of the number of lions (β_3 lions_ = 0.4, 95% CI = 0.09/0.70; Figure 1), with the probability of bunching being significantly higher for three versus one lion (Figure 2). For Pilanesberg elephants, on the other hand, the null model accounted for most of the weight (0.65), and model averaging did not reveal a significant difference in the probability of bunching for three versus one lion (β_3 lions_ = 0.16, 95% CI = −0.08/0.39; Figure 1 and Figure 2).

For the prolonged listening variable, the null model was the top model for both populations and accounted for 0.81 of the AICc weight for Pilanesberg (Table 1). For Amboseli, the second and third models still accounted for >0.30 of the AICc weight. For both Amboseli and Pilanesberg, the 95% confidence limits of the β-estimates slightly overlapped zero (β_3 lions_ = 0.24 (95% CI = −0.09/0.56) and 0.17 (95% CI = −0.12/0.46), respectively), but this was evident to a greater extent in Pilanesberg (Figure 1). Furthermore, there was less overlap in the standard error (SE) of the fitted values for Amboseli compared to Pilanesberg (Figure 2).

Meanwhile, for bunching intensity, the AICc weights were more evenly spread across four top models for the Amboseli dataset (Table 1). Model averaging revealed that the overall bunching intensity score for three roaring lions was significantly greater than for one lion (β_3 lions_ = 2.37, 95% CI = 0.33/4.41; Figure 1). There was also an increased probability of triggering a faster bunch for three roaring lions, whereas the observed bunching intensity for roars from one lion remained comparatively low across all three levels of response (Figure 3). For Pilanesberg, the top model accounted for 0.61 of the AICc weight and included the additive effect of number of lions and number of adult females (Table 1). Model averaging revealed that Pilanesberg matriarchs did not distinguish between one versus three lions (β_3 lions_ = 0.84, 95% CI = −0.46/2.14; Figure 1) nor did they significantly alter the strength of their bunching response to one versus three lions (Figure 3). See Appendix A for the model averaged parameters for each response variable in the top models.

## 4. Discussion

Our results demonstrated that elephant family groups in both study populations exhibited defensive bunching to playback experiments of lion roars, highlighting that lions were perceived as a legitimate threat by all family groups, regardless of developmental history or experience. In fact, the elephants in Pilanesberg exhibited a greater overall responsiveness to lion playbacks with a bunching probability of 57% compared to 40% in Amboseli. However, the family groups in Amboseli were able to reliably discriminate the greater predatory threat presented by three lions roaring versus one and tailor their bunching responses accordingly. This is evident from the greater probability of bunching and the significant increase in speed and cohesion of the bunch (bunching intensity) that was recorded following playbacks of three roaring lions. 

Bigger groups of hunting lions are invariably more successful in taking large-bodied prey such as elephants [18,19,35,36,38,39]. Discriminating between the numbers of roaring lions is—under normal circumstances—likely to be a universal skill among elephant family groups that is acquired at a relatively early stage of adulthood [4]. Nevertheless, the elephants in the Pilanesberg population did not show this behavioural distinction, with the ‘number of lions’ variable being non-significant across all of the analyses. Moreover, the intensity of bunching and, thus, the degree of preparedness to meet the predators, was very well tailored to the greater threat presented by three versus one lion in Amboseli but not in Pilanesberg. Interestingly, evidence of increased prolonged listening to the greater threat posed by three lions was limited for both populations. However, the elephants in Amboseli did exhibit a stronger relative listening response to three roaring lions compared with the elephants in Pilanesberg. It is noteworthy that prolonged listening was also a weaker behavioural response in our previous experiments exploring the threat assessment of elephants to different human groups [11], which may be due to the priority of coordinating a cohesive group reaction compared with gathering further information on the source of the danger. Indeed, when dealing with an imminent threat, such as an approaching predator, it is crucial that a fast and reliable assessment is made using the information that is immediately available. Our results indicate that the elephants in Amboseli seem to demonstrate much greater accuracy and consistency in decision making based solely on acoustic information, compared with the elephants in Pilanesberg. 

In cognitively advanced and highly social species, such as elephants, acquiring detailed knowledge about potential predators and making swift and accurate decisions is likely to involve social learning, particularly in environments where the levels and nature of threat are complex and dynamic across both time and space [50]. Indeed, experiencing these dangers first-hand could prove fatal. Experimental studies in mammals, birds and fish demonstrated that learning about predators from better-informed individuals is an effective strategy for acquiring knowledge that is critical for survival [51,52,53]. Although, to date, there are no studies that definitively demonstrate social learning among elephants, there is compelling evidence that knowledge transfer among conspecifics is a fundamental aspect of elephant society. For example, important information on nutritional food items appears to be passed from mother to her weaning calf [54] and from experienced to naive females on functionally appropriate oestrous behaviour [55], while young male elephants are likely to learn their crop-raiding strategies from older associates [56]. Elephants also proved capable of vocal imitation; a learnt form of communication that is believed to be very useful for maintaining individual social bonds in complex fission–fusion societies [57,58,59].

Although the elephants in Pilanesberg were able to associate lion roars with predatory threat, they did not exhibit the important, finer-scaled behavioural distinctions that would indicate that they were discriminating between the numbers of roaring lions, which pose different levels of danger. These skills could well involve a component of learning from older and more experienced females within the social group, particularly the matriarch [3,4]. When these animals are removed from the population (e.g., during culling operations), the surviving individuals no longer have the more experienced group members to learn from [26,29]. Evidence from playbacks of disturbed and aggressive bees to African elephants in Samburu National Reserve, Kenya, demonstrated that, while families typically retreated from the threat, a small sub-group that consisted of a 14-year-old female, her calf and a young male failed to respond [60]. It was suggested that these young elephants did not have the opportunity to either experience the considerable threat that bees represent, either directly or through social learning [60]. The same limitations in assessing the predatory threat presented by lion roars may apply to the family groups in Pilanesberg. 

Possessing detailed ecological knowledge can confer an advantage to long-lived and highly social species [4,5,61]. However, it appears that severe disruption of elephant social structure can compromise the quality of decision making when faced with assessment of fine-scale, ecological information (e.g., discriminating between sub-groups of predators). Similarly, when social knowledge was contrasted across these same two populations, the Pilanesberg elephants did not appear to use acoustic cues to distinguish between conspecifics on the basis of social identity or age-related dominance [29]. It is therefore plausible, that in profoundly socially disrupted populations such as Pilanesberg, the processing of essential social and ecological knowledge was affected both by experiencing severe trauma, which can directly impact neurological development [62], and by the loss of social learning opportunities due to the removal of older and more experienced individuals that are the repositories of this key knowledge [3,4]. As a result, the coordination and decision making of the matriarch and her family group could be compromised when faced with the assessment of threat situations, leading to potential fitness costs for individual elephants. Elephants in both our study populations showed a strong response to three lions roaring, but, in contrast to Pilanesberg, the behavioural reaction of the Amboseli elephants was greatly reduced when it came to the lower threat of one roaring lion. Ultimately, this tailored response is likely to reduce the energetic costs, lost feeding time and injury risks associated with overreacting. 

It is important to highlight the potential variation in lion predation risk experienced by elephants in different regions of Africa that we were unable to account for in our study. This is driven by the specific characteristics of the local lion populations, including average pride size, social stability and hunting preferences [18], all of which offer interesting avenues for future research. Furthermore, due to the challenges of conducting such an ambitious experimental study, we were only able to compare the behavioural responses in two elephant populations. Expanding this research to include more populations with different developmental histories, while also including playbacks of other predators (presenting different levels of risk), could produce a more robust framework for assessing the significance of the differences that we observed here and facilitate further insights into the threat assessment capability and ecological knowledge of this highly cognitive species. 

## 5. Conclusions

The findings presented here suggest that ecological knowledge is learned, retained, transmitted and disrupted in a similar manner to social knowledge, generating further evidence of the importance of maintaining population structure and integrity in highly social species. Social learning is likely to be a fundamental behavioural process by which crucial knowledge is passed from one individual to the next, enabling complex and informed decisions to be made in both ecological and social domains [50]. If populations experience extreme social trauma (e.g., through culling, poaching or translocation) then the mechanisms that allow effective knowledge acquisition and transfer between related individuals may be severely disrupted [29]. This can have long-term implications for population viability and effective conservation management, particularly in long-lived and cognitively advanced species (e.g., elephants, primates and cetaceans), which, through their ability to process complex social and ecological information, can maintain behavioural and cultural flexibility [10,15,16]. 

Recent research demonstrated that elephants orphaned because of poaching have a reduced probability of survival [33], while surviving adult females were shown to exhibit higher levels of stress hormones and lower reproductive output [31], which can have serious implications for population persistence. In addition to the more traditional focus on abundance, conservation practitioners also need to consider the significance of maintaining population social structure and function, as this has crucial fitness benefits for the population and can help buffer against external stressors that increasingly accompany anthropogenic disturbance [63,64]. Indeed, the findings from our study have significant implications for conservation interventions such as the reintroduction or supplementation of highly social species. 

Over recent decades, translocation has proved to be an effective conservation tool that has enabled populations of endangered species to be re-established in regions where the number of remaining individuals has become critically low or even extinct [65]. However, social functionality needs to be a central consideration for these management operations to maximise the chance of a successful outcome [66], with practitioners prioritising the crucial role of learned behaviours and knowledge transmission when deciding upon the age and social structure of the founding population. Our results add further weight to this argument, particularly for long-lived and cognitively advanced species. Future research could directly explore the links between the behavioural responses associated with detailed social and ecological knowledge acquisition and the potential fitness benefits (e.g., increased survival and/or reproductive output). Long-term studies of known individuals provide an excellent opportunity for further advances in our understanding of the critical role that knowledge acquisition, social learning and culture have in maintaining the population integrity of highly social and often endangered species. 

## Figures and Tables

**Figure 1 animals-12-00495-f001:**
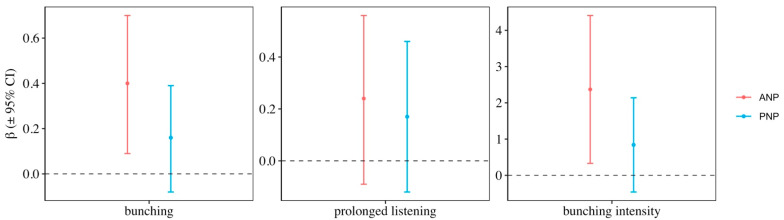
Model averaged β-estimates (±95%) of the number of lions parameter for Amboseli National Park (ANP) and Pilanesberg National Park (PNP). β-estimates were generated from the 6 model sets for each study population and were considered significant when the 95% CI did not overlap zero (dashed horizontal line).

**Figure 2 animals-12-00495-f002:**
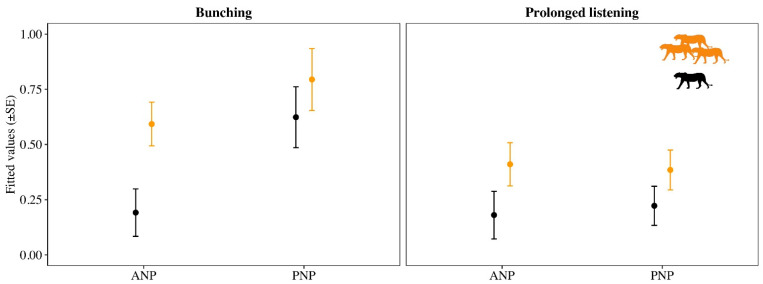
The probability of elephant bunching and prolonged listening during playbacks of one (black) versus three (orange) lions roaring in Amboseli National Park (ANP) and Pilanesberg National Park (PNP). Lion silhouette illustration was created by L.S.C.

**Figure 3 animals-12-00495-f003:**
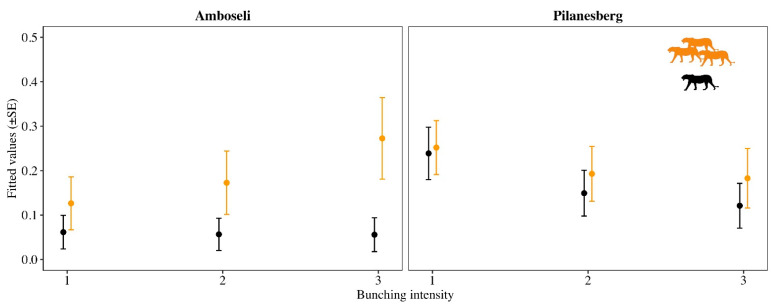
Relative difference in the bunching intensity metric for elephants receiving playbacks of one (black) versus three (orange) lions in Amboseli National Park and Pilanesberg National Park. Lion silhouette illustration was created by L.S.C.

**Table 1 animals-12-00495-t001:** Top models accounting for up to ≥0.95 AICc weight for bunching, prolonged listening and bunching intensity for the Amboseli (ANP) and Pilanesberg (PNP) populations. K is the parameter count for the model.

Response	Pop	Explanatory	K	AICc	ΔAICc	AICc Weight
Bunching	ANP	number of lions	4	60.2	0.00	0.85
		null	3	64.5	4.30	0.10
	PNP	null	3	81.4	0.00	0.65
		number of lions + number of adults	5	83.4	2.00	0.24
		number of lions	4	85.0	3.60	0.11
Prolonged listening	ANP	null	3	60.7	0.00	0.67
		number of lions	4	62.7	2.01	0.25
		number of lions + number of adults	5	65.2	4.44	0.07
	PNP	null	3	78.1	0.00	0.81
		number of lions	4	81.1	3.00	0.18
Bunching intensity	ANP	number of lions + number of adults	6	90.4	0.00	0.40
		number of lions	5	91.1	0.73	0.28
		number of lions + age of matriarch	6	91.9	1.50	0.19
		number of lions+ age of matriarch + number of adults	7	92.8	2.42	0.12
	PNP	number of lions + number of adults	6	141.7	0.00	0.61
		number of lions+ age of matriarch + number of adults	7	144.3	2.65	0.16
		null	4	144.9	3.24	0.12
		number of lions	5	145.8	4.16	0.08

## Data Availability

Data is contained within the Appendix A.

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
