# Peer review of "Social Disruption Impairs Predatory Threat Assessment in African Elephants"

_animals, 2022, doi:10.3390/ani12040495_

Round 1

Reviewer 1 Report

Overall:  Good paper and I have few comments that would improve upon the existing draft.  This paper is very relevant, especially considering the state of other wild and feral herds of social animals whose family and social lives may be significantly disrupted by human interventions. 

Abstract:  Good summary of knowledge in both the simple summary and the abstract.  Abstract provides sufficient knowledge about the study, history, and findings. 

Introduction:  Thorough and relevant history of both elephants and other large social mammals is explained with regard to social learning and stable social groups.  Specific history of elephants and the herds in question is also covered in sufficient detail.

Materials and Methods:

Very clear description of population, equipment, researchers and analysis.

Results:

Good use of visuals and tables to display results of findings. 

Discussion: 

Good discussion of how data and results fit into the existing literature with emphasis on previous literature on social learning (and lack thereof). 

Reviewer 2 Report

My only question concerns the use of Serengeti lion calls in the experiment. If prior research shows that elephants can detect different languages then I wondered if they could recognize and distinguish regional from non-local lion calls. While the Serengeti is geographically separated from Amboseli by many miles, I wondered about subtle regional differences in lion calls. This could influence the results. If this is so, then the South African Pilanesberg elephants might not have perceived the predation threat as intensively as those in Amboseli. 

Reviewer 3 Report

General Comments

This study asks the question of whether individual and group history plays a role in response to predator threats of differing levels. This is an interesting look at how anthropogenic influence may have long-term effects on anti-predator behavior through disruption of social groups. The writing was clear, and the methods were straightforward, and statistics seemed appropriate. I do have a two major concerns: the lack of independence within parks, and the statement of inability of differentiation for Pilanesberg elephants.

While impossible, I would have liked to have seen a few disrupted groups (i.e., early deaths of mothers for whatever reason such as natural disasters, poaching, etc.) in Amboseli and some more stable groups in Pilanesberg as the differences could also be differences in parks due to community composition, habitat, relationship/experience with lions, importance of different predators, etc. While there are many samples at each park, they are psudoreplicated as the samples from each park are not entirely independent.

I state this below, but this study cannot show that any of the elephants cannot differentiate as that is not the type of question that can be answered with this type of experiment. What these data show is that they do not behaviorally differentiate between one and three lions. Those are two very different things and the latter could occur while individuals can differentiate due to a variety of ecological and life-history reasons.

Specific Comments

51-53: why is this particularly relevant for the stated species compared to ones like passerines that live and travel in flocks? I would say that the need for information transmission between individuals is equally important. Maybe the type of information (i.e., long-term patterns, routes, etc.) might be different between these types of species, but information transfer between individuals is just as necessary for both large brained social species with overlapping generations and small brained social species with overlapping generations (e.g., learning about predators in many passerine species).

Do both groups have equal degrees of danger/threat from lions or are the lions controlled more in one park than the other to allow elephants to not be killed by them?

227: were they unable to make the distinction, or did they not show a distinction? I think that while they very well may not be able to make the distinction, their response only tells us that they do not behaviorally differentiate – they ‘do not show a distinction’. This is where it would be useful to have different family histories in both parks. There could be other legitimate reasons besides inability that families in one park do not respond differently to one vs. three lions but are still able to differentiate. They may simply view any lion threat the highest level of threat (i.e., three may be worse, but one is already so bad they max out their response), or increased aggression or trauma may cause more extreme defensive or aggressive responses to lower levels of danger (as suggested by more frequent bunching) so one lion gets as much response as three lions, though maybe one vs. five or seven may be a large enough distinction to get behavioral differences. Additionally, due to lion abundance, refuge abundance and availability, or personal history with lions individuals may be in just as much danger from one lion as many. I think the results say more about the threat the lions may pose across parks or individual sensitivity to danger than elephant perception of lion numbers.

278-280: however, in Pilanesberg, they responded to all lions much more like Amboseli responded to three lions, showing that they had a heightened response to one lion which could simply be a reflection of the perceived danger of one and three lions (i.e., maybe the same in Pilanesberg). If that is the case then their degree of preparedness was accurate as well.

293-296: this is also true of non-cognitively advanced social species (like social passerines) who may learn about predators and whose predators shift and change with migration, and season. I think that everything in this paper can be applied to any social species that uses social learning and stressing the ‘cognitively advanced’ aspect of elephants is not adding anything particularly strong to the argument – especially when “ . . . to date, there are no studies that definitively demonstrate social learning among elephants . . .”.

309-311: See the comment above regarding ‘inability’. Additionally, though, no other predators or predatory threat was tested so this statement is not supported (unless by sub-groups of predators that pose different levels of danger you specifically meant one vs three lions). If some experimental playbacks were hyenas or wild dogs and they didn’t respond differently to different threat predators, this statement would be accurate, but this was not done.

312-314: the question here is what might they learn? They obviously know the lions are dangerous, so do they then learn about how to modulate the response to danger (i.e., when they don’t need to respond to danger as much)?

Reviewer 4 Report

This study is uniquely positioned to contribute important insights into the impacts of anthropogenic disturbances on the ability of African elephant family units to accurately assess and respond appropriately to danger through the lenses of social learning and trauma. The future welfare of this highly social, cognitively-advanced, and long-lived species would benefit from the efforts of studies like this that could inform conservation practices and inspire future investigations on the topic.

To date, previous studies have predominantly focused on the direct impacts of extreme conditions, both natural and anthropogenic, on African elephants by measuring changes in population demographics, fecundity rates, and stress-induced responses (both physiological and behavioral). As such, I commend the authors’ efforts in collecting and presenting potential evidence for the indirect impacts of severe social disruption on knowledge transmission in ‘family groups’ decades after the founder population at Pilanesberg was formed. Acquiring important data such as those presented in this study is no small task, and I applaud the authors for all the time and hard work that went into conducting this research.

I also found the playback experimental design to be well thought out and takes into account potential habituation that may confound results. The criteria employed to evaluate appropriate responses to predation risk are behaviors well-documented elsewhere and are therefore a useful gauge to compare across populations. In addition, the comparative analyses between disrupted and stable populations builds on playback response research this group has previously published (McComb et al. 2011 and Shannon et al. 2013). The results showcase an intriguing difference between the responses made by family groups across populations, offering support for their conclusions on how anthropogenic disturbance effects ecological knowledge in African elephants at a finer-scale.

Overall, I found the topic intriguing and important, but there are some gaps in the study’s presentation and analysis that make it challenging to interpret the results in some places. Some critical components that would benefit from further elaboration and clarification are as follows: (1) contextual/background information pertaining to the two study populations (especially Pilanesberg and its founder population), lion predation pressures in both ecosystems, and descriptive data on the composition and age structure of matriarchs/family groups subjected to playback experiments; (2) clearer explanation of the statistical models employed and why two separate models were used and how the authors demonstrate that they are comparable, as well as how the results translate into probabilities that are compared across the two populations; and (3) further development in the discussion section on the topics of social trauma and long-term effects of severe disturbance in African elephants, how specifically this study informs conservation management and may be incorporated into practice, and the authors’ suggestions on ways others can build upon this body of research. By addressing these points, the authors would offer a stronger presentation of their valuable data. I recommend that this manuscript be accepted with major revisions outlined below.

Main Comments:

  1. Contextual/Background Information:
    1. Study Populations – A major point of framing throughout the manuscript pertains to the drastic differences in life experience and social structure between the Amboseli and Pilanesberg elephant populations, but this not fully supported by the information offered in text. I recommend the authors expand on the following: (1) the age range of the 76 founder orphans introduced to Pilanesberg (referred to as ‘young’ in line 141); (2) the age structure of the Pilanesberg population at the time of playback experiments; (3) a brief summary of the criteria used to estimate the age of elephants in Amboseli; and (4) a brief summary of the characteristics that define a “normal age and social structure” in Amboseli (line 121).
    2. Pressures of Lion Predation Across Regions – Given that this study tests behavioral responses to playbacks of lion roars, it suggests that the pressures of lion predation are similar across regions and that both elephant populations have similar experiences with lions. However, no information is provided on the presence or number of lions in both regions, nor any definitive evidence of lion predations on elephants at either field site. Stability of prides can vary across lion populations, particularly in relation to water availability and distance between waterholes. Hunting preferences may also factor into how elephants in these regions respond to an apparent threat. These factors should be addressed. In addition, it is recommended that the authors clarify what type of lion roars were utilized in playback experiments (e.g. contact call, territorial roars, etc.), as this may also affect the elephant response.
    3. Info on Matriarchs and Family Groups Presented with Playbacks – Given that the authors’ previous study demonstrated that older matriarchs were a key factor in determining behavioral responses to discrimination tasks (lines 157-159), the comparison of groups based on matriarchs of the same age class (lines 159-161), and the inclusion of ‘age of the matriarch’ as an additive variable in the analyses (Table 1), it would be important to provide a table on the ages of the matriarchs tested in both populations. This table would ideally include additional summary data on family group composition and age structure, group size, and, for at least Pilanesberg, relatedness (i.e. presence of genetic relatives or mother-offspring units). It would also be informative to include whether the Pilanesberg matriarchs (or others in the group) were one of the orphans that founded the population. If not all of the tested Pilanesberg matriarchs belonged to the original orphans, it would seem particularly relevant to incorporate this variable in the models to evaluate how this factor contributes to the resulting behavioral responses.
  1. Presentation of the Models and Results:
    1. The results needs to be clarified. For example, the procedure for transforming β-estimates (log-odds scale, though not stated in-text) to fitted values/probabilities is unclear and needs further explanation. In addition, a further explanation is needed as to the choice to employ separate models for these two systems and justify how they are comparable. I would also recommend that the authors provide a table outlining what the coefficients are, and even including an LaTeX formula outlining the β coefficients would aid the reader in discerning how the parameters and predictor variables were treated. Without these clarifications, it is difficult to assess the results and have confidence that the translation of the model outputs lead to significant results.
    2. The authors also state that preliminary analyses conducted in a previously published study showed that the presence of calves ≤3 years old had no effect on behavioral response (lines 226-229), but these analyses only take into account the Amboseli population. As such, a separate analysis for the Pilanesberg population would be needed for comparison and confirmation that similar patterns are observed across populations to validate one aspect of your modelling approach.
  1. Other Main Comments:
    1. There are a few key components missing from this paper that would offer more structure and clarity to the presentation of the study.
      1. Aims, predictions, and hypotheses – I would recommend adding a sentence or two in the last paragraph of the introduction that clearly summarizes the purpose and expectations of the study explicitly stating what is being compared.
      2. According to the Animals journal website, the conclusion section is mandatory. I would recommend adding a paragraph or two where you can elaborate on some of the future directions of this research, as well as how studies like these can better inform conservation management (please refer to comments above).
    2. Minor grammatical and structural errors should be corrected throughout. See recommendations below in line-by-line comments.
  1. Elaborating on Discussion Points:
    1. Social Trauma and Long-Term Consequences – A more comprehensive discussion would be very helpful, particularly on how social disturbance—such as poaching, culling practices, orphaning, and the loss of older, more knowledgeable leaders—potentially alters elephant social learning, with an emphasis on trauma and the duration of the effects observed in Pilanesberg (e.g. multiple decades at least). Given that the PNP population spent ~30 years in their new environment at the time of playback experiments, the formation of family groups that are more biologically-related and that are led by adult matriarchs would be expected. These older individuals certainly hold valuable knowledge and likely transmit information to conspecifics, but may have been stunted in their experiential development given the traumatic circumstances of their lives (i.e. culled families, translocation, and social structures built from young, unrelated individuals). The evidence presented in this paper indicates that this is indeed the case. In addition, the PNP population is fairly ‘young’ in comparison to the one in ANP, both in time spent at PNP and in terms of social development. As such, the cumulative repository of knowledge among PNP elephants is likely far more limited, and the transmission of this knowledge may be limited even further given the loss of core groups and individuals of close association (Archie et al. 2006; McComb et al. 2001) only 30 years prior. Further discussion on this point would be highly informative and an important motivator for this study and others that follow.
    2. Informing Conservation Management – In the abstract and discussion, the authors touch upon how understanding the impacts of anthropogenic disturbance on African elephants effects long-term population viability and informs conservation management practices for the species. I think the authors have offered a good demonstration of why these impacts are important, but how to apply them to conservation practice could be elaborated further and discussed with more specificity. Perhaps providing examples of management programs that have been effective at achieving/maintaining natural social structure and function would be a useful approach. Discussing programs that focus on managing other long-lived and cognitively advanced species could also offer valuable insights.
    3. Future Research Directions – A few recommendations and insights on future directions of this research would be of great interest and serve to aid others interested in filling gaps of knowledge on the topics discussed throughout this paper.

Line-by-Line Comments:

Line 33: The authors mention matriarchs of 50 years or older, but based on the previous statements about social disruption and lack of older individuals in the Pilanesberg population, I would have assumed there were no individuals of this age. Please clarify later in the introduction if this is just referring to the Amboseli population or if the Pilanesberg population have recovered a more natural social structure decades after disruption, yet still demonstrate a deficit in ecological knowledge and limited responses to different levels of threat.

Line 40-42: This sentence starts with, “this is likely to be exacerbated by…,” but in what way do these traumas further contribute to the abnormality of certain behavioral responses (i.e. directly through stress and negative effects on neurological development, indirectly through the loss of learning opportunities with older, more knowledgeable individuals, or both)?

Line 65: The “a” in Africana should be lower case.

Lines 62-76: It appears that there are two thoughts being presented here and they don’t quite connect. The first sentence is referring to cultural differences and behavioral variance across groups, populations, and regions, but is not well elaborated. The paragraph then brings up two solid examples in the literature pertaining to social learning, but this is not comparative in the way the first sentence suggests the example should be. Line 73-76 might work better at the beginning of this paragraph instead, and you could use another example of elephants from a different population instead of the dolphin one.

Lines 90-92: For an African elephant example, I recommend referring to Parker et al., Poaching of African elephants indirectly decreases population growth through lowered orphan survival, Current Biology (2021), https://doi.org/10.1016/j.cub.2021.06.091

Line 105: the authors reference Davidson et al. 2013 here, but I don’t believe this article mentions anything on group hunting strategies or the size of groups that were responsible for successfully killing elephant calves (just the difference in percent of kills across sex and season).

Line 106: please provide more contextual information about this previous study (i.e. field site), especially since you are comparing the results of two different populations in this current study.

Lines 134-136: Is this demographic information published somewhere? If so, please cite.

Line 138: Moss 2001 does an excellent job of spelling out exactly what techniques were employed when determining the approximate age of elephants born before 1972 in Amboseli, however authors should include the aging criteria used (i.e. by listing shoulder height, hindfoot length, etc.).

Line 141: I would recommend the authors remove the apostrophe quotes around the word ‘orphan,’ as it suggests that perhaps not all of the elephants in Pilanesberg were in fact orphans. If this is the case, please elaborate or adjust how you refer to the PNP founder elephants.

Lines 162-165: This is important information to include somewhere in the methods section (perhaps in Study Populations), but seems out of place in the playback section.

Lines 180-182: It would be helpful to clarify if you mean that repeat playbacks were matched to the first within a distinct playback session, but across playbacks sessions were varied or randomized. Is that what you are trying to say? If not, be sure to make this a bit clearer.

Line 184: You mention not testing family groups with calves younger than one month old. I would recommend adding a brief explanation for this, similar to the one presented in McComb et al. 2001 & 2011 which state how family groups respond with higher sensitivity to potential threats when young and vulnerable calves are present.

Lines 185-199: Please reference specific behaviors used for response criteria (i.e. defensive bunching and prolonged listening with other attentive behaviors like ears spread out, ears stiff, and head raised).

Possible relevant references:

  1. McComb et al. 2000 (doi:10.1006/anbe.2000.1406)
  2. McComb et al. 2011 (doi:10.1098/rspb.2011.0168)
  3. McComb et al. 2003 (doi:10.1006/anbe.2003.2047)

In addition, a table would serve you well in describing these criteria.

Lines 200-204: A clearer summary of the procedures and statistical analyses employed (i.e. the type of test run, statistical packages and software used, etc.), such as here with inter-rater reliability testing, helps back up your incredibly high irr results. Also, the authors should explain how videos were chosen for scoring by the independent rater (i.e. was it random?), as well as why 15% of the playbacks were chosen instead of a different proportion.

Line 217: What is meant by “the ability of elephant matriarchs to numerically assess lions from recordings.” I would suggest rephrasing to provide more clarity. Perhaps use the phrase “discriminate between different levels of lion predation risk,” since you have already established that the number of roars is a measure of this.

Line 218: For clarity, please define models and model sets. It would appear from these two paragraphs that a total of 36 models were used. Is this correct?

Lines 226 and Table 1: It is not clear what the null model represents, and how the model accounts for the difference in the number of lion roars presented. Does the null represent the single lion roar? Please clarify.

Lines 239, 242, and 250: In all of these lines, both Figures 1 and 2 are referenced when presenting results for the three lions β-estimates; however, it is not completely clear if both of these figures need to be referenced, or which figure is actually visualizing these results. For example, the values for the β-estimates and CIs presented in-text match those depicted in Figure 1, but is presented as β3 lions. Figure 1 does not indicate that these results are only showing model averaging for the three roars responses, but rather seems to suggest it showcases the average of all top models for each behavioral response category. Figure 2’s depiction is in fact demonstrating differences in responses towards three vs. one lion roar, but the values do not match those presented in thetext. Clarification and additional details in both captions and the main text would be most helpful, as well as in the data analysis section where model procedures and β-estimates are first introduced.

Line 265: I would recommend expanding the Figure 1 caption, given that the authors have acronym labels (e.g. ANP). I would also suggest the same when discussing confidence intervals, and would explain what the dashed lines for each graph represents (i.e. significance being determined by whether the CIs overlap 0, mentioned briefly in the main text). This would offer the reader more clarity without having to return back to the main text in order to find the information.

Line 268: I would clearly state that the yellow fitted values +/- SE represent the probability of responses towards three lions, whereas brown represents the probability of responses for one lion roar. The visual is attractive, but does not clearly act as a legend on its own without further explanation. These same comments also apply to Figure 3.

Line 277: Here, the authors have presented bunching probability percentages for each population (ANP = 40% and PNP = 57%), but these results have not been presented prior to this point. Please provide these values sooner and then reference them here again in the discussion.

Lines 316-317: This is an interesting piece of information, but is not entirely relevant to the topic (i.e. social learning in cognitively advanced species). As such, the clause pertaining to primates might be omitted for clarity. Further details and references on vocal mimicry in elephants and how it supports the social learning hypothesis would be ideal.

Possible references:

  1. Poole et al. 2005 (https://doi.org/10.1038/434455a)
  2. Stoeger et al. 2012 (DOI:10.1016/j.cub.2012.09.022)
  3. Stoeger and Manger 2014 (DOI:10.1016/j.conb.2014.07.001)

Lines 318-321: The authors mention that the Pilanesberg elephants were in fact able to associate lion roars with predation risk, but this is not well highlighted in the results. I would make it clearer in the introduction that the authors evaluated (1) whether Pilanesberg elephants were able to associate lion roars with predation risk, and (2) whether they were able to perform finer-scale discrimination. The focus was predominantly on the latter, but establishing and discussing this baseline capacity is important to elaborate on and include in the results section as well.

Lines 336-338: This could be made clearer: (1) PNP elephants were unable to distinguish caller identity or age differences given acoustic cues, and (2) age-related calls were simulated. I would at least mention that these were acoustic discrimination tasks. That way, the authors avoid making generalizations of the results presented in Shannon et al. 2013, since the ability to recognize conspecifics in other ways (visual, olfactory, etc.) was not explored in that particular study.

Lines 345-346: What results can be referred to here to confirm the strength of response to three lion roars? How is this strength measured, especially given the presentation of three behavioral response categories, and are all behavioral criteria taken into account when evaluating the strength of response to higher threat vs lower threat playbacks? Further clarification on this would allow for easier interpretation of the results and discussion points.

In addition, McComb et al. 2011 (doi:10.1098/rspb.2011.0168) graphically presents response probabilities in Fig. 1, 2b and 2d. I would recommend doing something like this, or perhaps a table of probability results. Also, if the analyses presented in this manuscript are based on raw playback data collected in Amboseli that have previously been presented in another publication (e.g. McComb et al. 2011), please state this clearly and reference.

Line 352: When referring to “what we know about social knowledge” in elephants, please cite the appropriate references. (i.e. Shannon et al. 2013, McComb et al 2001 & 2011 offer insights). Gobush et al. 2008 (doi:10.1111/j.1523-1739.2008.01035.x), focus more on the direct effects (e.g. reproductive outputs, mean group relatedness of disrupted populations, physiological stress) of anthropogenic disturbances on elephants, rather than the indirect effects presented here and in a few sources elsewhere (e.g. abnormal behavioral responses, given the lack of learning opportunities from older, more knowledgeable individuals). Emphasizing this knowledge gap would strengthen your manuscript, given that this study aims to help fill such a gap.

Round 2

Reviewer 3 Report

All of my questions/concerns have been addressed by the authors.

Author Response

We are grateful to the reviewer for their considered and detailed appraisal of our paper - and pleased to see that they are fully satisfied with the revised version and our responses to their comments.

Reviewer 4 Report

It is clear in this revision that the authors carefully considered the queries that I had about their manuscript. I appreciate all the additional information and modifications that the authors included in this revision (e.g. particularly the final paragraph of the introduction, the conclusion section, etc.). With regard to the methods used in their model, the explanations and clarifications were very helpful However, I have a few further queries, particularly in the methods section. Additional clarity was provided to my query about the model comparisons and null model, but that clarity is not included in the body of the manuscript. The query that I had was intended to help the reader understand what was done.  For example, additional information regarding the procedures employed to transform model odds and/or coefficients to probabilities (or fitted values in Fig. 2 & 3) would be helpful for the reader.

In addition, the authors supply supplemental tables of both the family herd demographics and the model coefficients, which is great, but the paper would benefit if this information was also supplied in the main text. For one, offering the table of coefficient values for top models as a main table would allow the readers to more easily evaluate the parameters that went into models that were averaged for beta-estimates. I also suggest below how the authors might consider summarizing the family group data as a main table. This would allow for a simple yet explicit comparison of family herds presented with lion roars across the two populations. Without this, it is challenging to assess whether other factors should have been included in the models.

Lastly, I can appreciate that some queries could be considered beyond the scope of this study. However, it would be helpful and informative for future researchers to have the authors briefly discuss possible factors that one might consider that could contribute to variation in behavioral responses for future studies, even if  beyond the scope of this study. For example, while the additional information provided about lion densities across regions is an improvement, other aspects of lion predation pressures (as noted in round 1) could also influence elephant behavioral responses. This might be an important factor for future researchers to consider (one example is how lions in some regions are known to be especially successful at hunting elephants, such as in the Savuti, and as such, elephants in these areas would be particularly wary).

Given the outlined minor additions above, I recommend that this manuscript be accepted with minor revisions.

Line-by-line Comments:

Line 68 – “a skill that must have involved learning the significance of local variants,” is an excellent addition to the previous sentence.

Line 69 – “bay” in the name “Shark Bay” should be capitalized.

Line 70 – “here mothers passing on this skill to their offspring via social learning,” appears to be a minor grammatical error. Perhaps “here” was meant to say “with?”

Line 129-131 – The last sentence of the final introduction paragraph is a great addition.

Line 149 – The authors have added the requested detail regarding the age of elephants upon introduction (<10 years), which is very helpful. However, even if a female released in 1981 was 10 years old and then was subjected to playbacks in 2010, that puts her at approximately 39 years of age. Based on this, why then was the <50 years old criteria used instead? Is this because ages were based on age classes, as defined in Moss 2001 (https://doi.org/10.1017/S0952836901001212)?

Lines 153-155 – Data on the composition and age structure of Pilanesberg family groups is mentioned here, but there is no citation, table, or figure listed. Where is this information available?

Lines 160-162 – The additional information on the density of lion populations is very helpful. I recommend adding a few sentences in the discussion that elaborates on how factors pertaining to lion pressures could be further explored and perhaps considered for model parameters in future studies. 

Line 166 – “giving elephant family groups playbacks…” is a bit awkward. Please consider rewording to something like, “broadcasting playbacks of three lions verses a single lion roaring to elephant family groups…”

Lines 142-145 & 172-175 – An abbreviated summary table in the main body of the paper of the data presented in Table S1,  including the mean/range of 1) matriarch age, 2) number of individuals per herd, 3) number of adults per herd, and 4) number of repeated playbacks per herd for each population would be very helpful. Having all of this information organized in one place and more easily accessible to the reader would improve clarity of the experimental design.

Line 214 – RE how videos were chosen randomly for independent rater scoring, further information on how the level of agreement between raters was assessed for bunching and prolonged listening would be important. Was this assessed with the spearman’s correlation coefficient as well? If so, I recommend adjusting the language slightly to clarify. If the spearman’s correlation coefficient was not used for this rater agreement assessment, please state what test was used.

Line 219-223 – How did the authors account for repeated measures (i.e. >1 playback exposures to a single herd)? Was a GLMM for repeated measures utilized or were scores averaged within herds? Added clarification on this point would be most appreciated.

Line 238 – An explanation of what the null model is and how it was used in this analysis was offered in the authors’ responses but should also be included in the paper. A brief additional note that states what the null model was used for (i.e. comparing model fit of single and additive variable models) would suffice. In addition, the authors state that both simple and additive models were compared directly to the null model (presumably a different null for each population). It would be helpful if the authors provide a bit more detail on the procedures that resulted in final modeled probabilities that were then used for direct comparisons between populations (see Line 242-246 comment).

Line 242 – Please write out “Akaike Information Criterion (AIC),” the first time it is introduced. Adding that AICc scores are corrected for small sample sizes would also be great.

Line 242-246 – While the authors explain how AICc weights were employed for both model comparisons and in choosing top models for model averaging, this section would benefit from a brief summary of the procedures used to calculate model probabilities. Given that both the beta-estimates and probabilities transformed from model outputs represents the study’s main results, additional information on these steps (even if brief) would clarify what was done. For example, it would be valuable to provide the computations and transformations (i.e. log-odds  odds  probabilities) performed prior to presentation of these results.

Line 249 – It is not clear how the overall probability of bunching was calculated due to the final clause in the statement referring to comparisons of the raw data. Was this overall probability not generated from the models in the same manner as the modeled probabilities of responses to one vs. three lion roars (as shown in Fig 2 & 3)?

Line 256 & 262 – It looks like the authors generally choose to present AICc weights in decimal form, but in places they are also offered as percents. One or the other should be chosen for consistency.

Line 284 – Figure 1’s caption is much clearer now.

Table 1 – I noticed that models with the explanatory variable of matriarch age did not explain a large proportion of the predictive power of the full model set, but this is not mentioned or discussed further. This result seems somewhat surprising, particularly for Amboseli elephants. It seems plausible that older matriarchs would possess more ecological/social knowledge than their younger counterparts, given that age and experience are connected. Do the authors have thoughts on this result? I would recommend something be mentioned on this point of interest, given how much is known about this population. Doesn’t this finding counter previous findings where matriarchs are thought to be repositories of knowledge, the older the wiser (McComb et al. 2011, DOI:10.1098/rspb.2011.0168)?

Line 395-396 – something appears to be missing: “the social and age structure of the founders needs to take into account the crucial role of learned behaviors…” I presume the last part of this sentence is referring to what conservation practitioners need to take into account, not the founders themselves.

Line 398-409 – The summary offered at the end is an excellent addition to the manuscript and provides an interesting discussion on future directions and how long-term monitoring enables optimal conservation management of cognitively advanced, social species like elephants.
